# Identifying near-optimal decisions in linear-in-parameter bandit models with continuous decision sets

**Sanjay P. Bhat**[1]                    **Chaitanya Amballa**[1]

[1]TCS Research, Hyderabad, India

## Abstract

We consider an online optimization problem in a bandit setting in which a learner chooses decisions from a continuous decision set at discrete decision epochs, and receives noisy rewards from the environment in response. While the noise samples are assumed to be independent and sub-Gaussian, the mean reward at each epoch is a fixed but unknown linear function of a feature vector, which depends on the decision through a known (and possibly nonlinear) feature map. We study the problem within the framework of best-arm identification with fixed confidence, and provide a template algorithm for approximately learning the optimal decision in a probably approximately correct (PAC) setting. More precisely, the template algorithm samples the decision space till a stopping condition is met, and returns a subset of decisions such that, with the required confidence, every element of the subset is approximately optimal for the unknown mean reward function. We provide a sample complexity bound for the template algorithm and then specialize it to the case where the mean-reward function is a univariate polynomial of a single decision variable. We provide an implementable algorithm for this case by explicitly instantiating all the steps in the template algorithm. Finally, we provide experimental results to demonstrate the efficacy of our algorithms.

## 1 INTRODUCTION

Multi-arm bandits have proved to be a fertile setting for studying various aspects of exploration and exploitation in sequential decision-making problems. While the regret minimization setting probes trade-offs between exploration and exploitation [Bubeck and Cesa-Bianchi, 2012], the pure exploration setting examines efficient exploration for maximizing information gain [Even-Dar et al., 2002, Bubeck et al., 2009]. Best arm identification (BAI) is one example of a pure exploration task where the learner seeks to identify the best arm through exploration. BAI is itself studied in two settings, namely, the fixed budget setting and the fixed confidence setting. In the fixed budget setting, the learner seeks to minimize the probability of misidentifying the optimal arm over a fixed number of trials [Audibert et al., 2010]. In contrast, in the fixed confidence setting, the aim of the learner is to minimize the number of trials needed to identify the optimal arm with a given level of confidence [Even-Dar et al., 2003].

Inclusion of additional structure in the reward environment adds a new dimension to the bandit problem. One structured bandit setting that has been widely considered in the literature is that of linear bandits. In a multi-arm linear bandit problem, each arm is associated with a feature vector in a finite-dimensional real vector space, and the mean reward of the arm is an unknown linear function of the feature vector. A more general version of the linear bandit problem results when the set of "arms" is a subset, not necessarily finite, of a real vector space. Unlike in the case of a multi-arm bandit where pulling one arm provides no information about another, the linear structure of the mean reward in a linear bandit problem opens up the possibility of learning optimal decisions even while sampling suboptimal ones.

The linear bandit problem has received significant attention in the regret minimization setting, both the case of finite arms as well as continuous decision sets [Auer, 2002, Dani et al., 2008, Agrawal and Goyal, 2013, Bartlett et al., 2008]. In contrast, the pure exploration setting for linear bandit problems has started gaining attention only relatively recently [Soare et al., 2014, Degenne et al., 2020, Garivier and Kaufmann, 2016, Yang and Tan, 2021, Karnin, 2016, Xu et al., 2018, Tao et al., 2018, Jedra and Proutiere, 2020]. What is more, except for the specific case of a spherical decision set considered in Jedra and Proutiere [2020], the literature on pure exploration in linear bandits has so far

*Accepted for the 38th Conference on Uncertainty in Artificial Intelligence* (UAI 2022).

focused on the case of finite decision sets only.

In this paper, we consider a bandit problem in which the mean reward is an unknown linear function of a feature vector that depends on the decision through a known, but possibly nonlinear, feature map. Furthermore, we do not assume the decision set to be finite. The motivation for our problem comes from real-life applications where the decision variable takes a large number of real values at a fine resolution, and the mean reward depends continuously on the decision variable. In such cases, it is more efficient to model the decision set as a continuum rather than a finite set. A prime example is that of dynamic pricing [Den Boer, 2015, Ganti et al., 2018, Keskin and Zeevi, 2014], where the seller of a product faces an unknown product demand that depends (possibly non-linearly) on the selling price of the product. The seller seeks to learn the selling price that results in the maximum revenue. In this case, it is common to model the selling price as a continuous variable and the revenue as a continuous function of the selling price. Additionally, approximating the revenue function as an unknown linear combination of a finite number of known basis functions yields a linear-in-parameter bandit model with a continuous decision variable.

While BAI algorithms in the finite arm case seek to find the best arm with high confidence, finding the best decision from a continuum of decisions can be prohibitively expensive. Hence we consider a $(\varepsilon, \delta)$-probably-approximately-correct (PAC) formulation, where the goal of the learner is to find a set of points which are $\varepsilon$-optimal with probability at least $1-\delta$. By building on the work of Soare et al. [2014], Jedra and Proutiere [2020], Kaufmann et al. [2016], we provide a lower bound on the sample complexity of $(\varepsilon, \delta)$-PAC algorithms in Section 3. Next, we use the notion of volumetric spanners [Hazan and Karnin, 2016] to devise VSBAI, a simple algorithm template for BAI in our setting in Section 4. We prove VSBAI to be $(\varepsilon, \delta)$-PAC, and provide upper bounds on its sample complexity.

In Section 5, we consider the case where the mean reward is a polynomial function of a single decision variable. We show that, in this case, a volumetric spanner can be computed using convex optimization, and indicate how the algorithm template VSBAI can be instantiated for BAI under polynomial rewards. Finally, we present experimental results in Section 6.

Before describing the problem setup in Section 2, we introduce some notation used throughout the paper. We use $\mathbb{R}$ and $\mathbb{Z}_+$ to denote the set of real numbers and positive integers, respectively, and $A^{\mathrm{T}}$ to denote the transpose of the matrix $A$. The 1-norm and 2-norm on $\mathbb{R}^n$ are denoted by $\|\cdot\|_1$ and $\|\cdot\|_2$, respectively. Given a function $g : \mathcal{D} \to \mathbb{R}$ and $\varepsilon > 0$, $s \in \mathcal{D}$ is $\varepsilon$-optimal for $g$ if $g(s') \leq g(s) + \varepsilon$ for all $s' \in \mathcal{D}$. A set $\mathcal{D}' \subseteq \mathcal{D}$ is $\varepsilon$-optimal for $g$ if every element of $\mathcal{D}'$ is $\varepsilon$-optimal for $g$. Finally, $\|g\|_\infty$ denotes the

sup norm of a real-valued function when its domain is clear from the context.

## 2 PROBLEM SETUP

We consider a bandit optimization setting in which a learner interacts with an environment at discrete decision epochs $t = 1, 2, \ldots$. At each period $t \in \mathbb{Z}_+$, the learner chooses a decision $s_t$ from a compact decision set $\mathcal{D} \in \mathbb{R}^d$ and receives a noisy reward $y_t = \mu^{\mathrm{T}} x_t + \eta_t$, where the $f$-dimensional feature vector $x_t = \phi(s_t)$ is related to the decision $s_t$ through a continuous feature map $\phi : \mathcal{D} \to \mathbb{R}^f$, $\mu \in \mathbb{R}^f$ is a parameter vector, and $\{\eta_t\}_{t \in \mathbb{Z}_+}$ is a noise sequence. Our reward model is thus given by

$$y_t = g_\mu(s_t) + \eta_t, \ t \in \mathbb{Z}_+, \tag{1}$$

where, for each $\theta \in \mathbb{R}^f$, $g_\theta : \mathcal{D} \to \mathbb{R}$ is defined by $g_\theta(s) = \theta^{\mathrm{T}} \phi(s)$.

Without any real loss of generality, we assume that $\phi(\mathcal{D})$ is not contained in any proper linear subspace of $\mathbb{R}^f$. In addition, our results make use of one or the other of the following two assumptions on the noise sequence.

*Assumption* 1. The noise sequence $\{\eta_t\}_{t \in \mathbb{Z}_+}$ is a sequence of zero mean i.i.d $\sigma$-sub Gaussian random variables for some $\sigma > 0$. Specifically, for each $i \in \mathbb{Z}_+, \eta_i$ satisfies $\mathbb{E}(e^{t\eta_i}) \leq e^{\frac{\sigma^2 t^2}{2}}$ for all $t \in \mathbb{R}$.

*Assumption* 2. The noise sequence $\{\eta_t\}_{t \in \mathbb{Z}_+}$ is a sequence of zero mean i.i.d. Gaussian random variables with variance $\sigma^2$ for some $\sigma > 0$.

Note that Assumption 2 is a special case of Assumption 1. We assume that, in the case of either of the two assumptions above, the learner knows $\sigma$. In addition, she also has access to the feature map $\phi$. However, the parameter vector $\mu$ is unknown to the learner.

In a best-arm identification setting, the learner's goal is to identify a maximizer $s^*$ of $g_\mu$ by using the observations $\{(s_i, y_i)\}_{i=1}^T$ collected over a decision horizon $T$. However, the presence of noise makes it impossible to identify an optimizer with certainty over a finite horizon. Hence it is standard practice in the literature to seek an algorithm that returns a set that contains the desired optimizer to a high level of confidence. Such an algorithm typically comprises of a sampling rule $\pi$ that determines the decision $s_t \in \mathcal{D}$ to explore at time $t$ given the history of observations up to $t - 1$, a stopping rule that decides if the exploration conducted so far is sufficient, and an estimation rule that computes a set that contains the desired optimizer to a high level of confidence. We make these ideas more precise in the next section.

# 3 $(\varepsilon, \delta)$-PAC ALGORITHMS AND THEIR SAMPLE COMPLEXITY

Complexity lower bounds on algorithms for best arm identification in a PAC setting have been studied before for the case where the decision set is finite Soare et al. [2014], Jedra and Proutiere [2020], Degenne et al. [2020], Kaufmann et al. [2016], Xu et al. [2018]. While the analysis we present below follows similar ideas, the continuous nature of the decision set makes it necessary to formally define the elements mentioned above using a little more machinery.

To this end, we note that a sampling rule could also make use of internal randomization in addition to the past history of decisions and rewards. It is easy to see that any randomization scheme requiring $n$ random variables at each decision epoch can be implemented using $n$ i.i.d. samples of a random variable uniformly distributed on the unit interval. Hence, to represent a general sampling rule more formally, we consider the Cartesian product $\mathcal{S} \stackrel{\text{def}}{=} \mathcal{D} \times \mathbb{R} \times [0,1]^n$, where $n \geq 0$ is a fixed integer. $\mathcal{S}$ is the set of triplets of decision, reward, and a set of $n$ auxiliary quantities used for internal randomization. For each $t$, we denote by $\Omega_t$ the set of sequences in $\mathcal{S}$ of length $t$, and by $\Omega$ the set of all infinite sequences in $\mathcal{S}$. We use $h_t = \{(s_i, y_i, u_i)\}_{i=1}^t$ to denote a general sequence in $\Omega_t$. We assume that $\mathcal{D}$ is a Borel set. By forming products of the Borel $\sigma$-algebras of $\mathcal{D}$, $\mathbb{R}$ and $[0,1]^n$, we obtain a $\sigma$-algebra $\mathcal{F}_t$ on $\Omega_t$ for each $t$, as well as a $\sigma$-algebra $\mathcal{F}$ on $\Omega$. Moreover, on letting $\mathcal{F}_0$ denote the trivial $\sigma$-algebra on $\Omega$, we obtain a filtration $\{\mathcal{F}_t\}_{t=0}^{\infty}$ on $\Omega$.

Next, we define a sampling rule $\pi$ to be a sequence $\{\pi_t\}_{t \in \mathbb{Z}_+}$ along with a Borel measure $\lambda$ on $[0,1]^n$, where $\pi_1$ is a stochastic kernel on $\mathcal{D}$ given $[0,1]^n$ and, for each $t > 1$, $\pi_t$ is a stochastic kernel on $\mathcal{D}$ given $\Omega_{t-1}$ and $[0,1]^n$. In other words, for each $t > 1$, the following holds: for each $h_{t-1} \in \Omega_{t-1}$ and $u \in [0,1]^n$, $\pi_t(\cdot|h_{t-1}, u)$ is a measure on the Borel $\sigma$-algebra of $\mathcal{D}$, while for each Borel subset $A$ of $\mathcal{D}$, $\pi_t(A|\cdot, \cdot)$ is a Borel-measurable function on $\Omega_{t-1} \times [0,1]^n$. Informally speaking, $\lambda$ is the measure used to sample an element of $[0,1]^n$ for any internal randomization used by the sampling rule while, for every $t > 1$, the measure $\pi_t(\cdot|h_{t-1}, u)$ describes the conditional distribution of the decision sampled at time $t$ given the history $h_{t-1} \in \Omega_{t-1}$ up to time $t-1$ and the randomly sampled $u \in [0,1]^n$. A similar interpretation applies for $t = 1$.

Any algorithm used by the learner can be represented by the tuple $\mathcal{A} = (n, \lambda, \pi, \tau, \mathbb{F})$, where $n$, $\lambda$ and $\pi$ are as described above, $\tau$ is a stopping time with respect to the filtration $\{\mathcal{F}_t\}_{t=0}^{\infty}$ representing the stopping condition of the algorithm, and $\mathbb{F}$ is a set-valued map that maps each finite history in $\Omega$ to a subset of $\mathcal{D}$. The algorithm terminates at the random time $\tau$ and returns the set $\mathbb{F}(h_\tau)$ upon terminating.

It is natural to represent the environment as a stochastic kernel $Q^\mu$ on $\mathbb{R}$ given $\mathcal{D}$, such that the measure on $\mathbb{R}$ given by $Q^\mu(\cdot|s)$ describes the conditional distribution of the reward (1) given the decision $s \in \mathcal{D}$. The interaction between the algorithm and the environment induces Borel measures $\mathbb{P}^{\mathcal{A},\mu}$ on $\Omega$ and $\mathbb{P}_t^{\mathcal{A},\mu}$ on $\Omega_t$ for each $t \in \mathbb{Z}_+$ (see Proposition 7.28 of Bertsekas and Shreve [1996]).

Finally, given $\varepsilon > 0$ and $\zeta \in \mathbb{R}^f$, we let $\mathcal{O}_\varepsilon(\zeta) \subseteq \mathcal{D}$ denote the set of decisions that are $\varepsilon$-optimal for the function $g_\zeta$. We seek an algorithm $(n, \lambda, \pi, \tau, \mathbb{F})$ such that, given $\varepsilon > 0$ and $\delta \in (0,1)$, the set $\mathbb{F}(h_\tau)$ returned by the algorithm on termination is $\varepsilon$-optimal for $g_\mu$ and contains the true optimal decision together with probability at least $1 - \delta$. We make this class of algorithms more precise in the next definition.

**Definition 3.1.** Given $\varepsilon > 0$ and $\delta \in (0,1)$, an algorithm $\mathcal{A} = (n, \lambda, \pi, \tau, \mathbb{F})$ is $(\varepsilon, \delta)$-PAC for the environment (1) if the stopping time $\tau$ is finite $\mathbb{P}^{\mathcal{A},\mu}$-almost-surely and $\mathbb{P}^{\mathcal{A},\mu}(\{\arg\max_{s \in \mathcal{D}} g_\mu(s) \subseteq \mathbb{F}(h_\tau) \subseteq \mathcal{O}_\varepsilon(\mu)\}) \geq 1 - \delta$.

The expected sample complexity of an algorithm $\mathcal{A} = (n, \lambda, \pi, \tau, \mathbb{F})$ is the expected number of decisions explored by the algorithm till termination, and is simply given by $\mathbb{E}^{\mathcal{A},\mu}(\tau)$, where $\mathbb{E}^{\mathcal{A},\mu}(\cdot)$ denotes expectation under $\mathbb{P}^{\mathcal{A},\mu}$. Next, we provide a lower bound for the expected sample complexity of a $(\varepsilon, \delta)$-PAC algorithm. To do so, we need one more notation. Given $\zeta \in \mathbb{R}^f$ and $\varepsilon > 0$, the $\varepsilon$-*alternative of* $\zeta$ is the set $\text{Alt}_\varepsilon(\zeta) = \{\zeta' \in \mathbb{R}^f : \mathcal{O}_\varepsilon(\zeta) \cap \mathcal{O}_\varepsilon(\zeta') = \varnothing\}$. We are now ready to state our lower bound. The proof, which builds on ideas given in Soare et al. [2014], Jedra and Proutiere [2020], Kaufmann et al. [2016], is given in Appendix A in the supplementary material.

**Theorem 3.2.** *Suppose Assumption 2 holds. Let $\varepsilon > 0$ and $\delta \in (0,1)$, and suppose $\mathcal{A} = (n, \lambda, \pi, \tau, \mathbb{F})$ is a $(\varepsilon, \delta)$-PAC algorithm for (1). Then*

$$\mathbb{E}^{\mathcal{A},\mu}(\tau) \geq \frac{2\sigma^2 \ln\left(\frac{1}{2.4\delta}\right)}{\inf\limits_{\zeta \in \text{Alt}_\varepsilon(\mu)} \|g_\mu - g_\zeta\|_\infty}. \tag{2}$$

# 4 VSBAI: AN ALGORITHM TEMPLATE

In this section, we present VSBAI, a general template for an $(\varepsilon, \delta)$-PAC algorithm for the bandit optimization problem described in Section 2, and provide a sample complexity bound for it. We prefer to use the term template rather than an algorithm as some of the steps of the template can only be implemented if $\mathcal{D}$ and $\phi$ are specified.

VSBAI combines two ideas, namely,

1. obtain a $\varepsilon$-optimal set for $g_\mu$ from a uniform approximation for $g_\mu$, and

2. with high probability, obtain a uniform approximation of $g_\mu$ by regressing the rewards obtained for decisions sampled at points of a suitable exploration basis for $\mathcal{D}$.

We elaborate on each of these two aforementioned ideas next.

## 4.1 APPROXIMATE OPTIMIZERS FROM UNIFORM APPROXIMATIONS

The intuition behind the first idea listed above is illustrated in Figure 1 for the case where $d = 1$. The thick solid curve in the figure depicts the graph of a uniform approximation $\hat{q}$ of an unknown function $q$ represented by the thin solid curve. Suppose the uniform approximation error does not exceed $\frac{\varepsilon}{4}$ for some $\varepsilon > 0$, that is, $\|q - \hat{q}\|_\infty < \frac{\varepsilon}{4}$ holds. The two dashed curves are graphs of the functions $\hat{q} \pm \frac{\varepsilon}{4}$, which serve as upper and lower bounds on the unknown function $q$. In other words, the graph of $q$ must lie within the region bounded by the two dashed curves. In the figure, the approximation $\hat{q}$ achieves its maximum at $\hat{s}$, while $s^*$ is the maximizer of $q$. The horizontal segment shown in the figure represents a set $\mathcal{D}'$ such that the approximation $\hat{q}$ does not fall below its maximum value $\hat{q}(\hat{s})$ by more than $\frac{\varepsilon}{2}$ on $\mathcal{D}'$. One can intuitively see from the figure that the set $\mathcal{D}'$ must contain the maximizer $s^*$ of the unknown function $q$. Moreover, the absolute difference between the values of the unknown function $q$ at any two points in the set $\mathcal{D}'$ cannot exceed the difference $\varepsilon$ between the maximum and minimum values on $\mathcal{D}'$ of the upper and lower dashed curve, respectively. In other words, the set $\mathcal{D}'$ is $\varepsilon$-optimal for $q$.

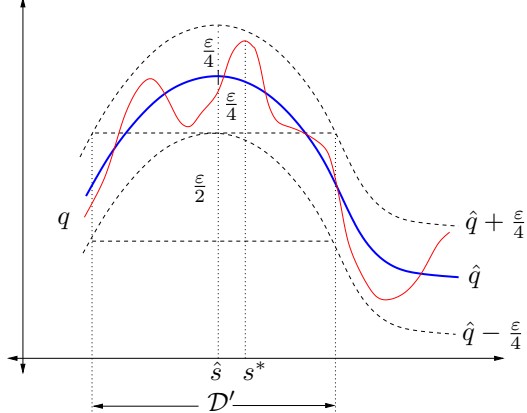

Figure 1: Obtaining $\varepsilon$-optimal points for $q$ from its uniform $\frac{\varepsilon}{4}$-approximation $\hat{q}$

The next proposition formalizes the intuition reflected in Figure 1. The proof is given in Appendix B in the supplementary material.

**Proposition 4.1.** *Let $\varepsilon > 0$, and suppose $q, \hat{q} : \mathcal{D} \to \mathbb{R}$ are such that $\|\hat{q} - q\|_\infty \leq \frac{\varepsilon}{4}$. Let $\hat{s} \in \arg\max_{s \in \mathcal{D}} \hat{q}(s)$. Then the set $\mathcal{D}' \stackrel{\text{def}}{=} \{s \in \mathcal{D} : \hat{q}(s) \geq \hat{q}(\hat{s}) - \frac{\varepsilon}{2}\}$ is $\varepsilon$-optimal for $q$, and contains $\arg\max_{s \in \mathcal{D}} q(s)$.*

## 4.2 UNIFORM APPROXIMATION OF THE REWARD FUNCTION

The Cauchy-Schwarz inequality gives $|g_{\hat{\mu}}(s) - g_\mu(s)| \leq \|\phi(s)\|_2 \|\hat{\mu} - \mu\|_2$ for every $s \in \mathcal{D}$ and $\hat{\mu} \in \mathbb{R}^f$. The compactness of $\mathcal{D}$ now shows that any estimate $\hat{\mu}$ of $\mu$ yields an uniform approximation $g_{\hat{\mu}}$ of $g_\mu$. Hence, an obvious and popular means of obtaining an approximation to the unknown reward function $g_\mu$ is to estimate $\mu$ from observed decisions and rewards using least-squares regression. It is not surprising, therefore, that either ordinary least squares (OLS) or regularised least squares forms a part of almost every algorithm available for linear bandit problems in the stochastic as well as adversarial settings with finite arms or continuous arms. We briefly review OLS before proceeding.

At the end of $t$ decision epochs, the learner has access to observations $\{(x_i, y_i)\}_{i=1}^t$, where $x_i = \phi(s_i)$ is the feature vector of the $i$th decision $s_i$, and $y_i$ is the corresponding observed reward. Letting $X_t \stackrel{\text{def}}{=} [x_1, \ldots, x_t] \in \mathbb{R}^{f \times t}$ and $y^t \stackrel{\text{def}}{=} [y_1, \ldots, y_t]^{\mathrm{T}} \in \mathbb{R}^t$, the OLS estimate $\hat{\mu}_t$ of $\mu$, based on the data $\{(x_i, y_i)\}_{i=1}^t$, is obtained by solving $\min_{\hat{\mu} \in \mathbb{R}^f} \|y^t - X_t^{\mathrm{T}} \hat{\mu}\|_2^2$, and is given by

$$\hat{\mu}_t = (X_t X_t^{\mathrm{T}})^{-1} X_t y^t. \tag{3}$$

For deriving (3), it is assumed that $X_t$ has rank $f$, which necessarily implies that $t \geq f$.

The parameter error $\hat{\mu}_t - \mu$ clearly depends on the choice of the decisions $s_1, \ldots, s_t$. Indeed, on letting $\eta^t = [\eta_1, \ldots, \eta_t]^{\mathrm{T}}$ denote the vector of noise samples till time $t$, it is easy to use (1) and (3) to show that

$$\hat{\mu}_t - \mu = (X_t X_t^{\mathrm{T}})^{-1} X_t \eta^t. \tag{4}$$

In a regret minimization setting, the decisions need to be chosen in an adaptive manner so that the required trade-off between exploration and exploitation can be achieved. Even in the pure exploration setting of best-arm identification in a finite multi-arm bandit problem, decisions have to be adaptive so that the exploration budget is diverted away from arms as and when they are revealed to be sub-optimal, since exploring one arm gives no information about another arm. In contrast, in the pure exploration setting that we are considering for the linear bandit problem, each decision that improves the estimate of $\mu$ also improves the accuracy of the approximation of $g_\mu$ over the whole decision domain $\mathcal{D}$. This suggests the possibility of using non-adaptive (that is, deterministic) sampling of the decision space for the purpose of constructing an OLS-based approximation of $g_\mu$.

In this case, it is natural to consider a *volumetric spanner* as a low variance exploration basis (as defined in Hazan and Karnin [2016]) for sampling the reward function. We review the necessary background next.

## 4.3 VOLUMETRIC SPANNERS

Suppose $L > 0$ and $m \geq f$. A $(L, m)$-*volumetric spanner* for $\phi(\mathcal{D}) \subseteq \mathbb{R}^f$ is a subset $\{x_1, \ldots, x_m\}$ of $\phi(\mathcal{D})$ such that, for every $z \in \mathcal{D}$, there exists $c_1, \ldots, c_m \in \mathbb{R}$ satisfying $z = c_1 x_1 + \cdots + c_m x_m$ and $c_1^2 + \cdots + c_m^2 \leq L^2$. Recall that, for every $z \in \mathbb{R}^f$ and $X \in \mathbb{R}^{f \times m}$ with $m \geq f$, $c = X^{\mathrm{T}}(XX^{\mathrm{T}})^{-1}z$ is the minimum-2-norm solution of the equation $Xc = z$. Hence, it follows from the definition that, if $\{x_1, \ldots x_m\}$ is a $(L, m)$-volumetric spanner for $\phi(\mathcal{D}) \subseteq \mathbb{R}^f$, then $\|X^{\mathrm{T}}(XX^{\mathrm{T}})^{-1}z\|_2 \leq L$ for all $z \in \phi(\mathcal{D})$, where $X = [x_1, \ldots, x_m] \in \mathbb{R}^{f \times m}$. In particular, if $m = f$, then $\|X^{-1}z\|_2 \leq L$ for every $z \in \phi(\mathcal{D})$. The last observation implies that there exists no $(L, f)$-volumetric spanner for $\phi(\mathcal{D})$ for $L < 1$. A $(1, m)$-volumetric spanner was called a volumetric spanner in Hazan and Karnin [2016] irrespective of $m$. Since the cardinality of the volumetric spanner will be required in our algorithm, we choose not to suppress it.

It will be convenient to define $p_1, \ldots, p_m \in \mathcal{D}$ to be $(L, m)$-*volumetric points* for the pair $(\phi, \mathcal{D})$ if $\{\phi(p_1), \ldots, \phi(p_m)\}$ is a $(L, m)$-volumetric spanner for $\phi(\mathcal{D})$.

Since a volumetric spanner forms a critical component of the algorithm that we present in the next subsection, it is important to consider the existence of such spanners as well as algorithms for computing them. We start with an easy observation. Our assumption that the set $\phi(\mathcal{D})$ of feature vectors is not contained in a proper linear subspace of $\mathbb{R}^f$ implies that $\phi(\mathcal{D})$ contains a set of $f$ linearly independent vectors. Since $\phi(\mathcal{D})$ is compact, it is easy to see that any linearly independent subset of cardinality $f$ will serve as a $(L, f)$ volumetric spanner for sufficiently large $L$. It is separately known that, being compact, $\phi(\mathcal{D})$ possesses a $(1, m)$-volumetric spanner for some $m \leq 12f$. In addition, if $\phi(\mathcal{D})$ is finite, then the aforementioned volumetric spanner can be constructed in polynomial time (see Theorem 3 of Hazan and Karnin [2016] for both facts above as well as additional details). We will see later that the sample complexity of our algorithm grows as $L^2 m$, and can be improved if a $(L, m)$-volumetric spanner with lower values of $L$ and $m$ is chosen. It is easy to see that the union of two $(L, m)$-volumetric spanners yields a $(L/\sqrt{2}, 2m)$-volumetric spanner, indicating that it is possible to reduce $L$ by considering volumetric spanners with more elements. In this context, the following bound proved in Appendix B in the supplementary material is of interest.

**Lemma 4.2.** *If $L > 0$ and $m \geq f$ are such that there exists a $(L, m)$-volumetric spanner for $\phi(\mathcal{D})$, then $L^2 m \geq f$.*

The lower bound in Lemma 4.2 is achieved by a $(1, f)$-volumetric spanner.

## 4.4 VSBAI: DESCRIPTION AND ANALYSIS

The template algorithm VSBAI that we present requires a set of $(L, m)$-volumetric points $\{p_1, \ldots, p_m\}$ for the pair $(\phi, \mathcal{D})$, for some $L \geq 1$ and $m \geq f$. The algorithm proceeds in rounds with each round consisting of $m$ decision epochs. In each round, the algorithm picks the points $\{p_1, \ldots, p_m\}$ in sequence as the decisions for that round. In the notation of section 3, the template algorithm is given by a tuple $\mathcal{A}^* = (n^*, \lambda^*, \pi^*, \tau^*, \mathbb{F}^*)$ whose sampling rule $\pi^*$ is defined by

$$\pi_t^*(\cdot | u) = \delta_{p_i}(\cdot), \ i = 1 + (t \bmod m), \quad (5)$$

for every $t \in \mathbb{Z}_+$ and $u \in [0, 1]^{n^*}$, where $\delta_s(\cdot)$ denotes the Dirac measure at $s \in \mathcal{D}$. Note that the sampling rule $\pi^*$ is deterministic, and hence the choices of $n^*$ and $\lambda^*$ are immaterial.

As described in subsection 4.2, our template algorithm $\mathcal{A}^*$ involves obtaining successively better uniform approximations of $g_\mu$ using a sequence of OLS estimates of $\mu$ obtained through (3). Our next result gives a high probability bound on the uniform error with which the estimate $g_{\hat{\mu}_{km}}$ obtained after $k$ rounds approximates $g_\mu$. The proof is given in Appendix C in the supplementary material.

**Proposition 4.3.** *Consider an algorithm $\mathcal{A}^*$ whose sampling rule is described by (5). Let $k \in \mathbb{Z}_+$ and $\varepsilon > 0$, and suppose Assumption 1 holds. Then*

$$\mathbb{P}^{\mathcal{A}^*, \mu}(\|g_{\hat{\mu}_{km}} - g_\mu\|_\infty > \varepsilon) \leq \beta\left(k, \frac{\varepsilon}{L}\right), \quad (6)$$

*where*

$$\beta(k, \varepsilon) \overset{\text{def}}{=} 2^{\frac{f}{2}} \exp\left(-\frac{k\varepsilon^2}{4\sigma^2}\right). \quad (7)$$

Propositions 4.1 and 4.3 immediately suggest the stopping criterion that yields an $(\varepsilon, \delta)$-PAC algorithm under the sampling rule described by (5). Indeed, by Proposition 4.3, choosing

$$\tau^* = \inf\left\{km : \beta\left(k, \frac{\varepsilon}{4L}\right) < \delta\right\} \quad (8)$$

ensures that, with probability at least $1 - \delta$, the uniform approximation condition required by Proposition 4.1 holds with $q = g_\mu$ and $\hat{q} = g_{\hat{\mu}_{\tau^*}}$. Letting $\mathcal{D}_{\tau^*} = \mathbb{F}(h_{\tau^*})$ to be the set $\mathcal{D}'$ in Proposition 4.1 then ensures that $\mathcal{D}_{\tau^*}$ is $\varepsilon$-optimal for $g_\mu$ with the same probability. The resulting algorithm is given as Algorithm 1 below.

In Algorithm 1, $\beta$ is taken to be given by (7). Also, the steps at lines 10 and 12 in the algorithm come from (20) in the supplementary material.

The main result of this section given below states that VSBAI is $(\varepsilon, \delta)$-PAC.

**Algorithm 1** VSBAI

1: **Input:** $\varepsilon > 0$, $\delta \in (0, 1)$, sub-Gaussianity parameter $\sigma$, $(L, m)$-volumetric points $p_1, \ldots, p_m$ for $(\phi, \mathcal{D})$
2: Set $B_{L,m} = [\phi(p_1), \ldots, \phi(p_m)]$
3: Initialize $k \leftarrow 1$, $r \leftarrow 0$
4: Set STOP = False
5: **while** STOP == False **do**
6:      Initialize reward vector $\bar{y}^k = []$
7:      **for** $t = 1, \ldots, m$, **do**
8:          Apply decision $s_{(k-1)m+t} \leftarrow p_t$
9:          Observe reward $y_{(k-1)m+t}$
10:          Augment reward vector
          $\bar{y}^k \leftarrow [(\bar{y}^k)^{\mathrm{T}}; y_{(k-1)f+t}]^{\mathrm{T}}$
11:      **end for**
12:      Update total reward vector $r \leftarrow r + \bar{y}^k$
13:      **if** $\beta(k, \frac{\varepsilon}{4L}) < \delta$ **then**
14:          STOP = True
15:      **else**
16:          $k = k + 1$
17:      **end if**
18: **end while**
19: $\tau^* \leftarrow km$
20: $\hat{\mu}_{\tau^*} \leftarrow \frac{1}{k}(B_{L,m}B_{L,m}^{\mathrm{T}})^{-1}B_{L,m}r$
21: Pick $\hat{s} \in \arg\max_{s \in \mathcal{D}} g_{\hat{\mu}_{\tau^*}}(s)$.
22: $\mathcal{D}_{\tau^*} = \{s \in \mathcal{D} : g_{\hat{\mu}_{\tau^*}}(s) \geq g_{\hat{\mu}_{\tau^*}}(\hat{s}) - \frac{\varepsilon}{2}\}$
23: **Output:** $\mathcal{D}_{\tau^*}$

---

**Theorem 4.4.** *Suppose Assumption 1 holds. Then Algorithm 1 terminates in at most $\tau^* \leq m[1 - 64L^2\sigma^2\varepsilon^{-2}\ln(2^{-\frac{f}{2}}\delta)]$ decision epochs. Furthermore, with $\mathbb{P}^{\mathcal{A}^*,\mu}$-probability at least $1 - \delta$, the set $\mathcal{D}_{\tau^*}$ returned by the algorithm is $\varepsilon$-optimal for $g_\mu$ and contains all the maximizers of $g_\mu$. In particular, Algorithm 1 is $(\varepsilon, \delta)$-PAC.*

*Proof.* Let $\tau^*$ be as computed by Algorithm 1, and let $k = \tau^*/m$. The upper bound for $\tau^*$ comes from using (7) in the stopping condition (8). Next, consider the event $\mathcal{E} = \{\|g_{\hat{\mu}_\tau} - g_\mu\|_\infty > \frac{\varepsilon}{4}\}$. By Proposition 4.3 and the definition (8) of $\tau^*$, it follows that $\mathbb{P}^{\mathcal{A}^*,\mu}(\mathcal{E}) < \beta(k, \frac{\varepsilon}{4L}) < \delta$. Proposition 4.1 now implies that, on the complement of the event $\mathcal{E}$, $\mathcal{D}_{\tau^*}$ is $\varepsilon$-optimal for $g_\mu$ and contains all the maximizers of $g_\mu$. This completes the proof. $\square$

Note that three critical steps in the algorithm depend on the pair $(\phi, \mathcal{D})$, namely, computation of the $(L, m)$ volumetric points used as inputs to the algorithm, computation of an optimizer $\hat{s}$ for the approximation $g_{\hat{\mu}_{\tau^*}}$ at line 21, and computation of the set $\mathcal{D}_{\tau^*}$ at line 22 of the algorithm. Hence we view the algorithm more as a template requiring the three aforementioned steps to be worked out for specific problem instances. We present a simple example considered in Jedra and Proutiere [2020] to illustrate these steps.

## 4.5 LINEAR BANDIT ON THE UNIT SPHERE

Let $f > 1$, and choose $\mathcal{D}$ to be the unit sphere $\mathrm{S}^{f-1} \overset{\text{def}}{=} \{s \in \mathbb{R}^f : \|s\|_2 = 1\}$. Let $\phi : \mathrm{S}^{f-1} \to \mathbb{R}^f$ be the inclusion map. Then the reward function in (1) becomes $g_\mu(s) = \mu^{\mathrm{T}}s$.

Any set of $f$ orthonormal vectors is seen to be a set of $(1, f)$-volumetric points for the pair $(\phi, \mathrm{S}^{f-1})$. For every non-zero $\theta \in \mathbb{R}^f$, $\arg\max_{s \in \mathrm{S}^{f-1}} g_\theta(s)$ equals $\{\|\theta\|_2^{-1}\theta\}$. Line 21 of Algorithm 1 thus returns $\hat{s} = \|\hat{\mu}_{\tau^*}\|_2^{-1}\hat{\mu}_{\tau^*}$, while the set $\mathcal{D}_{\tau^*}$ at line 22 of Algorithm 1 is given by the "spherical cap" $\{s \in \mathrm{S}^{f-1} : \hat{s}^{\mathrm{T}}s \geq 1 - \frac{\varepsilon}{2\|\hat{\mu}_\tau\|_2}\}$.

Under Assumption 2, Theorem 4 of Jedra and Proutiere [2020] gives a lower bound for the sample complexity of any $(\varepsilon, \delta)$-PAC algorithm $\mathcal{A}$ for the case of the unit sphere considered here. On using inequality (3) of Kaufmann et al. [2016], the lower bound given by Jedra and Proutiere [2020] may be written as

$$\mathbb{E}^{\mathcal{A},\mu}(\tau) \geq \frac{\sigma^2(f-1)}{20\varepsilon\|\mu\|_2} \ln\left(\frac{1}{2.4\delta}\right) \tag{9}$$

for $\varepsilon < \|\mu\|_2/5$. Jedra and Proutiere [2020] also provide an algorithm for this case, and show that the sample complexity of their algorithm recovers the dependence on $\varepsilon$, $f$ and $\delta$ seen in the lower bound (9) asymptotically as $\delta \to 0$ (see Theorem 5 of Jedra and Proutiere [2020]). Interestingly, the sampling rule given by Jedra and Proutiere [2020] for their algorithm involves choosing $f$ orthogonal vectors in a round-robin manner just as mentioned above. However, their stopping rule is more intricate.

On using $L = 1$ and $m = f$, the upper bound provided by Theorem 4.4 for Algorithm 1 under Assumption 2 reduces to

$$\tau^* \leq f\left[1 + \frac{64\sigma^2}{\varepsilon^2}\ln\left(\frac{2^{\frac{f}{2}}}{\delta}\right)\right]. \tag{10}$$

On comparing (9) and (10), we see that while the dependence of the sample complexity of Algorithm 1 on $\delta$ compares favourably with the lower bound (9), the dependence on $\varepsilon$ does not, at least for small values of $\varepsilon$. This could indicate that either the lower bound is conservative (for $\delta > 0$), or that Algorithm 1 is sub-optimal. Closing this gap remains an open problem.

Before proceeding, we comment on the possible reason for the suboptimality of VSBAI in relation to the sample complexity lower bound (9), as well as the difference in the sample complexities of VSBAI and the algorithm of Jedra and Proutiere [2020]. As mentioned above, while the sampling rule used in both algorithms is the same, the stopping rules are different. The stopping rule in Jedra and Proutiere [2020] is designed to stop the exploration as soon as the accumulated data is sufficient to confidently distinguish the true linear function from the closest linear function that has

a completely different set of approximate optimizers (that is, functions corresponding to parameter vectors from the so called alternative set). In contrast, the stopping rule in VSBAI stops the exploration only when, with high probability, the true linear function is approximated sufficiently well uniformly everywhere by the OLS estimate without any reference to the alternative set. We believe that this difference in the nature of the stopping rules is the reason for both, the superiority of the asymptotic sample complexity (as $\delta \to 0$) of the algorithm of Jedra and Proutiere [2020] over that of VSBAI, as well as the suboptimality of VSBAI. We add, however, that the stopping rule from Jedra and Proutiere [2020] requires solving an optimization problem at every decision epoch, and is therefore difficult to implement.

It is easy to see from Theorem 4.4 that the best sample complexity for Algorithm 1 results when $L = 1$ and $m = f$, that is, when a set of $(1, f)$-volumetric points is available for the pair $(\phi, \mathcal{D})$. The unit sphere example considered in this subsection provided a simple setting in which a set of $(1, f)$-volumetric points is available. In the next section, we will see a nontrivial setting where such a set of volumetric points exists, and can be computed easily.

## 5 UNIVARIATE DECISION VARIABLE WITH POLYNOMIAL REWARD

As a concrete instance of the general problem setup described in Section 2, we consider the case where the reward function $g_\mu$ in (1) is a univariate polynomial of degree $f - 1 > 0$ on an interval $[p_{\min}, p_{\max}] \subset \mathbb{R}$ for some $p_{\max} > p_{\min}$. To cast this case of polynomial rewards in our general setup, we let $\mathcal{D} \overset{\text{def}}{=} [p_{\min}, p_{\max}]$ and define $\phi : [p_{\min}, p_{\max}] \to \mathbb{R}^f$ by $\phi(s) \overset{\text{def}}{=} [1, s, \ldots, s^{f-1}]^{\text{T}}$. Then, for each $\theta \in \mathbb{R}^f$, $g_\theta$ is the univariate polynomial in $s$ of degree $f-1$ with coefficients given by the parameter vector $\theta$. Our next result shows that a set of $(1, f)$ volumetric points for the pair $(\phi, \mathcal{D})$ exists. The proof is given in Appendix D in the supplementary material.

**Proposition 5.1.** *Suppose $p_{\min} \leq p_1 \leq \cdots \leq p_f \leq p_{\max}$. Then the following two statements are equivalent.*

1. *The points $p_1, \ldots, p_f \in \mathcal{D}$ are $(1, f)$ volumetric points for the pair $(\phi, \mathcal{D})$.*

2. *The points $p_1, \ldots, p_f$ satisfy $p_{\min} = p_1 < p_2 \cdots < p_f = p_{\max}$ and*

$$\sum_{1 \leq j \leq f, j \neq i} \frac{1}{p_i - p_j} = 0, \quad i = 2, \ldots, f-1. \quad (11)$$

Equations (11) also appear in Amballa et al. [2021], where it is shown that (11) provide necessary and sufficient conditions for the points $\phi(p_1), \ldots, \phi(p_f)$ to form a barycentric spanner for the set $\phi(\mathcal{D})$. Proposition 5.1 above thus

implies that, in the case of univariate polynomial reward functions, a barycentric spanner is also a volumetric spanner. Amballa et al. [2021] also show that the equations (11) possess a unique solution, and this solution may be computed efficiently either by numerically solving the algebraic equations (11) or by solving a convex optimization problem. Furthermore, volumetric points for the general case $\mathcal{D} = [p_{\min}, p_{\max}]$ can be easily recovered from volumetric points for the special case $\mathcal{D} = [0, 1]$. This means that, effectively, the solution of (11) needs to be computed just once for a given $f$.

Proposition 5.1 enables us to implement the initialisation step on line 1 of Algorithm 1. The optimization $\arg \max_{s \in \mathcal{D}} g_{\hat{\mu}_{\tau^*}}(s)$ at line 21 of the algorithm may be performed by finding the roots of the derivative of the polynomial $g_{\hat{\mu}_{\tau^*}}$ and picking the maximizer of $g_{\hat{\mu}_{\tau^*}}$ among them by evaluation. Note that the set $\mathcal{D}_{\tau^*}$ at line 22 may be a disjoint union of multiple closed intervals. The endpoints of these intervals may be found by numerically computing roots of the polynomial $s \mapsto g_{\hat{\mu}_{\tau^*}}(s) - g_{\hat{\mu}_{\tau^*}}(\hat{s}) + \frac{\varepsilon}{2}$. A sequence of easy checks can then be used to pair the roots to yield the actual intervals whose union equals $\mathcal{D}_{\tau^*}$. Thus, VSBAI can be implemented rather easily for the case where the mean reward is a polynomial function of a single decision variable. The algorithm VSBAI-Poly in Appendix E of the supplementary material provides an instantiation of VSBAI for the case of polynomial rewards and a single decision variable.

## 6 EXPERIMENTAL RESULTS

In this section we present experiments comparing VSBAI with other recent algorithms in various settings described below. We first consider the toy example considered in Fiez et al. [2019] and Jedra and Proutiere [2020], and compare the sample complexities along with run-times in different scenarios. We also present in Appendix G in the supplementary material some experimental results for the polynomial setting described in Section 5. All the results that we present were computed on an AMD Ryzen 5 2500U CPU with Radeon Vega mobile gfx × 8 with 12GB memory.

### 6.1 MULTI-ARM SETTING

We consider the "finitely many arms with moderate gaps" example first presented in Fiez et al. [2019] and further used in Jedra and Proutiere [2020]. The decision set is a finite collection of $n$ 2-dimensional unit vectors given by $\mathcal{D} = \{[0, 1]^{\text{T}}, [\cos(3\pi/4), \sin(\pi/4)]^{\text{T}}\} \cup \{[\cos(\pi/4 + \phi_i), \sin(\pi/4 + \phi_i)]^{\text{T}} : i = 3, \ldots, n\}$, where $n \geq 3$. Each choice of the angles $\{\phi_i\}_{i=3}^n$ represents a problem instance. In order to examine robustness across different problem instances, our experiments involve randomly sampling sets of these angles to generate different problem instances. The

results we present below use $\mathcal{N}(0, .09)$ for generating the angles $\{\phi_i\}_{i=3}^n$. We also report the results from using the uniform distribution on the interval $[0, 0.1]$ in Appendix G in the supplementary material. Typical arm configurations obtained by sampling the angles are depicted in figures 3 and 4 in Appendix F (see the supplementary material).

The feature map $\phi$ is taken to be the identity map, and the reward is given by (1) with $\mu = [1, 0]^{\mathrm{T}}$. Also, Assumption 2 holds with $\sigma = 1$. To implement VS-BAI on a problem instance, we first find the index $j$ of the arm which has the least inner product with the arm $[\cos(3\pi/4), \sin(\pi/4)]^{\mathrm{T}}$. We then find a value of $L$ such that the arms $[\cos(3\pi/4), \sin(\pi/4)]^{\mathrm{T}}$ and $j$ form a set of $(L, 2)$-volumetric points for the decision set. These $(L, 2)$ volumetric points are used to initialize VSBAI, which is run with $\varepsilon = 0.1$ and $\delta = 0.05$.

For drawing a comparison, we consider the LAZYTS (averaged) algorithm proposed in algorithm 1 in Jedra and Proutiere [2020], the RAGE algorithm given as algorithm 1 in Fiez et al. [2019], and the ORACLE algorithm given by equations (4) and (5) of Soare et al. [2014]. For each choice of the size of the decision set, we generate 20 instances of the problem by sampling as many sets of the angles using either the normal distribution or the uniform distribution as described above. In addition to comparing sample complexities, we also compare run-times as a measure of efficiency. The results that we present below for sample complexity and run time were obtained by averaging these quantities over all 20 problem instances for each algorithm.

Table 1 gives the sample complexities of the three baselines along with the VSBAI algorithm as the number of arms increase. We observe that, for all the baselines, the sample complexity grows with the number of arms, but the sample complexity of VSBAI remains almost constant. This is not surprising. While the other algorithms need to know the number of arms, VSBAI is independent of the number of arms. We also note the standard deviation (over the randomly generated problem instances) of the sample complexity for VSBAI decreases as the number of arms increase. In contrast, it increases for the baselines. This can be explained by observing that the value of $L$ used by VS-BAI can be expected to be closer to 1 as the number of arms increase.

In Table 2, we present the run-times of the algorithms compared in Table 1. As in Table 1, the run-times are averaged over 20 problem instances. We note that VSBAI takes roughly a constant time to terminate whereas the run time of all the other algorithms increases as the number of arms increase. This is because all the three baselines attempt to find the best arm among all the arms. As a consequence, they can end up sampling the best two arms a large number of times in a scenario where the best two arms are very close to each other. VSBAI does not suffer from this draw-

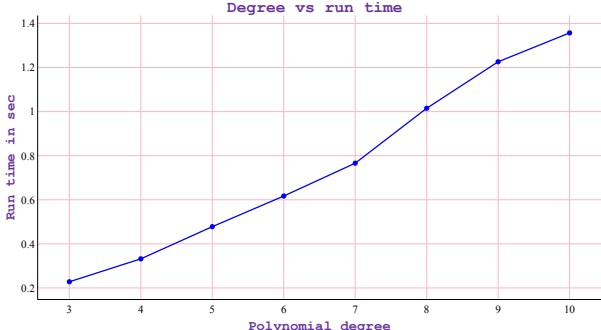

Figure 2: Run-time of VSBAI for polynomial reward functions

back as it seeks to find the best arm only to a certain degree of approximation, and this is a task that does not increase in difficulty with the number of arms. Also, in situations where the run-time is of importance, VSBAI makes it possible to use $\varepsilon$ as an additional tuning parameter to balance accuracy and speed.

## 6.2 POLYNOMIAL SETTING

Next, we present results for the case of polynomial rewards considered in section 5 with the decision set chosen to be the interval $[1, 10]$. As described in that section, the algorithm template VSBAI specializes to VSBAI-Poly, which is given as Algorithm 2 in Appendix E (see the supplementary material).

To implement VSBAI-Poly, we computed $(1, f)$-volumetric points for this problem using (11) of Proposition 5.1 and the numerical technique suggested in Amballa et al. [2021]. We ran the VSBAI-Poly for various degrees ranging from 3 to 10. Although the noise sequence used was Gaussian (that is, satisfying Assumption 2) with $\sigma = 10$, VSBAI-Poly was run using the sub-Gaussian tail bound (7). The error tolerance $\varepsilon$ was fixed to be 6 while the confidence parameter $\delta$ was chosen to be 0.1.

Figure 2 represents the run-time of VSBAI-Poly as the degree of the polynomial increases. The plot shows the run time averaged over 20 polynomials all having their maximum values around 350, but otherwise chosen randomly. As expected, the run time increases with the degree.

## 7 CONCLUSION

We have considered a bandit problem in which the mean reward is a linearly parametrized (but possibly nonlinear) function on a continuous decision set. We have used a $(\varepsilon, \delta)$-PAC formulation in which the goal is to find a set of points that are $\varepsilon$-optimal with probability at least $1 - \delta$. We have given a lower bound on the sample complexity

| Algorithm | LazyTS | | Rage | | Oracle | | VSBAI | |
|---|---|---|---|---|---|---|---|---|
| No. of Arms | Mean | Std | Mean | Std | Mean | Std | Mean | Std |
| 10 | 3490.05 | 1121.99 | 7617.4 | 2989.33 | 3470.05 | 1102.36 | 48919.8 | 487.87 |
| 20 | 72081.1 | 65078.96 | 103903.1 | 85734.65 | 47876.4 | 41692.63 | 48075.9 | 226.94 |
| 100 | 146331.55 | 64260.81 | 623143.05 | 366464.09 | 217162.25 | 111605.07 | 47381.3 | 44.41 |
| 1000 | 1218591.27 | 39881.14 | 16235680.31 | 5974249.14 | 7500331.73 | 2882866.47 | 47239.8 | 3.87 |

Table 1: Average sample complexity for the setting described in subsection 6.1

| Algorithm | LazyTS | | Rage | | Oracle | | VSBAI | |
|---|---|---|---|---|---|---|---|---|
| No. of Arms | Mean | Std | Mean | Std | Mean | Std | Mean | Std |
| 10 | 1.75 | 0.48 | 0.27 | 0.05 | 0.01 | 0 | 1.46 | 0.04 |
| 20 | 26.79 | 23.58 | 0.81 | 0.2 | 0.2 | 0.18 | 1.38 | 0.03 |
| 100 | 63.38 | 27.19 | 2.34 | 0.3 | 2.02 | 0.86 | 1.44 | 0.04 |
| 1000 | 39141.2 | 1270.31 | 120.92 | 6.01 | 116.56 | 38.37 | 1.4 | 0.03 |

Table 2: Run-time in seconds for the setting described in subsection 6.1

of $(\varepsilon, \delta)$-PAC algorithms. We have used the notion of volumetric spanners to devise a simple $(\varepsilon, \delta)$-PAC algorithm template and provided an upper bound on its sample complexity. As a special case of our general setting, we have also considered the case where the mean reward is a polynomial function of a single decision variable, and indicated how all the problem-specific steps in VSBAI can be instantiated to apply to this case. VSBAI showed advantages in experiments in terms of run time and sampling complexity when compared to recent algorithms proposed for the BAI problem in linear bandits with finite arms.

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
