# OpenReview forum: "Identifying near-optimal decisions in linear-in-parameter bandit models with continuous decision sets"
_auai.org/UAI/2022/Conference — UAI 2022 Poster_

### Official Review · Reviewer_71aX · 2022-04-01

**Q2(1) Originality/Novelty:** 2
**Q2(2) Significance/Impact:** 2
**Q2(3) Correctness/Technical Quality:** 3
**Q2(6) Clarity Of Writing:** 3
**Q6 Overall Score:** 7
**Q8 Confidence In Your Score:** 1

**Q1 Summary And Contributions:**

The paper deals with online optimization of the linear bandit problem with continuous decision sets and noisy rewards from the environment. The authors proposes an algorithm template to find a near optimal decision. Unlike most works in the close literature, the proposed template is not restricted to finite sets but can be applied to infinite ones using sample techniques. Finally, they provide both theoretical and experimental results to justify the method and show its efficiency.

**Q2 Assessment Of The Paper:**

More detailed information regarding each of these aspects is given below:

**Q2(4) Quality Of Experiments (Optional):**

4: Excellent: The experimental evaluation is comprehensive and the results are compelling.

**Q2(5) Reproducibility:**

4: Excellent: Key resources (e.g., proofs, code, data) are available and key details (e.g., proof sketches, experimental setup) are comprehensively described for competent researchers to confidently and easily reproduce the main results.

**Q3 Main Strengths:**

The paper is well written and organized.

The paper tackles an interesting and well-studied problem by providing a more general approach  as it can deal with larger decision sets comparing to the literature.

The proposed approach is justified both theoretically and empirically.

**Q4 Main Weakness:**

In my opinion, the main weakness of the paper is its accessibility. The paper is very technical, which is very interesting for an expert researcher in the field. For an non expert, the paper is a bit hard to read.

**Q5 Detailed Comments To The Authors:**

I am not an expert in continuous linear bandit, so maybe its a naive question, but I wonder if the infinite continuous decision set is a theoretical framework, or is there any application to this setting? Since the main contribution is to deal with infinite continuous sets, it would be interesting to highlight the relevance of such sets in your context.

Minor remark :

page1 in the abstract: May be you could specify somewhere the meaning of PAC.

At multiple places : you could use \mathbb{N} instead of \mathbb{Z}^+

page 3 section 4: VSBAI combinines two ideas → VSBAI combines two ideas

**Q7 Justification For Your Score:**

The interest of the subject and the quality of the paper: it is well written and it gives both theoretical and experimental justification of the proposed method.

**Q9 Complying With Reviewing Instructions:**

1: Yes.

---

### Official Review · Reviewer_q1As · 2022-04-09

**Q2(1) Originality/Novelty:** 3
**Q2(2) Significance/Impact:** 2
**Q2(3) Correctness/Technical Quality:** 3
**Q2(6) Clarity Of Writing:** 4
**Q6 Overall Score:** 7
**Q8 Confidence In Your Score:** 2

**Q1 Summary And Contributions:**

This paper considers a bandit problem (Best Arm Identification) with arms are defined in a continuous decision set. In particular, mean reward is assumed to be linearly parametrized with known (nonlinear) feature map. The authors devised a template algorithm based on a volumetric spanner in a PAC setting and provided its sample complexity bound.



**Q2 Assessment Of The Paper:**

More detailed information regarding each of these aspects is given below:

**Q2(4) Quality Of Experiments (Optional):**

3: Good: The experimental evaluation is adequate, and the results convincingly support the main claims.

**Q2(5) Reproducibility:**

3: Good: Key resources (e.g., proofs, code, data) are available and key details (e.g., proofs, experimental setup) are sufficiently well-described for competent researchers to confidently reproduce the main results.

**Q3 Main Strengths:**

While a similar setting was considered in the previous literature, the continuous nature of the decision set makes the analysis of complexity lower bounds non-trivial for the template algorithm. This paper provides a template algorithm and rigorous analysis for the said algorithm.

**Q4 Main Weakness:**

It would be reasonable to consider a real-world environment where the problem setting is motivated. Bandit literature focus on different structural assumptions and how they can be exploited in the learning algorithm. However, the paper neither contains an appealing motivational example in introduction nor considers empirical evaluation based on a real-world setting.

Whether it is "weakness" can be subjective given that the bandit literature sometimes focused more on "theory" as applications can be found in the future...

**Q5 Detailed Comments To The Authors:**

Since I am not an expert on deriving sample complexity, I focused on the presentation and flow. It is fairly understandable for bandit paper the notation-heavy nature of the paper. Yet, the authors tried to best to make paper readable by providing detailed explanations so that the readers can follow smoothly.



**Q7 Justification For Your Score:**

strength (algorithm and theoretical analysis) far outweighs the weakness (which may not be essential or somewhat orthogonal to the contribution of the paper per se.)

**Q9 Complying With Reviewing Instructions:**

1: Yes.

---

### Official Review · Reviewer_NRay · 2022-04-14

**Q2(1) Originality/Novelty:** 2
**Q2(2) Significance/Impact:** 3
**Q2(3) Correctness/Technical Quality:** 3
**Q2(6) Clarity Of Writing:** 3
**Q6 Overall Score:** 6
**Q8 Confidence In Your Score:** 3

**Q1 Summary And Contributions:**

This paper studies best arm identification with fixed confidence in linear bandits without restricting the decision set to be finite. A lower bound on the sample complexity of $(\epsilon, \delta)$-PAC algorithm is provided. A template for called $(\epsilon, \delta)$-PAC algorithm is proposed which can be implemented if the decision set and the feature map are specified. An upper bound on its sample complexity is provided with experimental results validating the theoretical guarantees.

**Q2 Assessment Of The Paper:**

More detailed information regarding each of these aspects is given below:

**Q2(4) Quality Of Experiments (Optional):**

2: Fair: The experimental evaluation is weak: important baselines are missing, or the results do not adequately support the main claims.

**Q2(5) Reproducibility:**

2: Fair: Key resources (e.g., proofs, code, data) are unavailable but key details (e.g., proof sketches, experimental setup) are sufficiently well-described for an expert to confidently reproduce the main results.

**Q3 Main Strengths:**

1. This paper considers a problem setting which is unexplored in the literature.

2. The mathematical treatment is thorough and is presented in an easy-to-follow manner.


**Q4 Main Weakness:**

1. The code is unavailable.
2. Their ideas could be described in more detail, especially section 4.1 and section 4.4

**Q5 Detailed Comments To The Authors:**

1. As mentioned in the answer to Q4, please consider expanding section 4.1 and section 4.4 as I believe they are central to explaining your proposed approach. I understand that due to space constraints, this could be challenging. Perhaps some of the non-essential details from Section 6 could be moved to the supplementary material to make some space?

2. Please consider adding a likely explanation for the gap between the lower bound and the upper bound on sample complexity for small $\epsilon$.

**Q7 Justification For Your Score:**

The work appears to be incremental although non-trivial.

**Q9 Complying With Reviewing Instructions:**

1: Yes.

---

### Official Review · Reviewer_zerJ · 2022-04-17

**Q2(1) Originality/Novelty:** 3
**Q2(2) Significance/Impact:** 3
**Q2(3) Correctness/Technical Quality:** 3
**Q2(6) Clarity Of Writing:** 3
**Q6 Overall Score:** 7
**Q8 Confidence In Your Score:** 3

**Q1 Summary And Contributions:**

The paper studies the sample complexity of the best-arm-identification for linear stochastic bandits when the action set is a subset of R^d. As the action set is a continuum, the core contribution of the paper is the required measure-theoretic formulation to describe the problem, and the topological definitions of the closest different arm - for example the denominator in Theorem 3.2. The paper gives a general algorithm recipe which it shows to be computable in a special 1dim case.

**Q2 Assessment Of The Paper:**

More detailed information regarding each of these aspects is given below:

**Q2(4) Quality Of Experiments (Optional):**

3: Good: The experimental evaluation is adequate, and the results convincingly support the main claims.

**Q2(5) Reproducibility:**

3: Good: Key resources (e.g., proofs, code, data) are available and key details (e.g., proofs, experimental setup) are sufficiently well-described for competent researchers to confidently reproduce the main results.

**Q3 Main Strengths:**

The key contributions of this paper

1. Formulate the Best arm identification for general continuum armed bandits. Prior work had defined and produced algorithms for the special case of the action set being the unit sphere.

2. Showing that a non-adaptive algorithm based on volumetric spanners is competitive to achieve the best-arm identification goal.

3. In the case of unit-sphere, the proposed non-adaptive algorithm is optimal with respect to $\delta$, but not $\varepsilon$. This is an intriguing open problem for the community.


**Q4 Main Weakness:**

The write-up comparing the present work with that of Jedra and Proutiere can be improved.

In particular, this was my inference from the paper. Jedra and Proutiere give an asymptotically optimal algorithm for the unit sphere. However the proposed algorithm although is universal, but is sub-optimal in the case of unit-sphere - in which case Jedra and Proutiere have a tailored algorithm.

Is this inference correct ? If so, I would encourage the authors to state it more explicitly and provide intuition if possible on why the algorithm of Jedra and Proutiere achieves superior performance for this special case.

**Q5 Detailed Comments To The Authors:**

See above.

**Q7 Justification For Your Score:**

The positives are nice in this paper. The negatives are mostly in the writing that I believe can be fixed in the writing.

**Q9 Complying With Reviewing Instructions:**

1: Yes.

---

### Decision · Program_Chairs · 2022-05-15

**Decision:**

Accept (Poster)

**Comment:**

Meta Review: This paper studies best-arm identification (BAI) in the fixed-confidence setting with a continuous decision set. This is important since many BAI algorithms are elimination-based and assume a discrete decision set. The challenge of the continuous setting is that the set of non-eliminated hypotheses needs to be represented compactly. The reviews of the paper did not raise any major issues, except that the paper may be too technical to be accessible to a broader audience. Similarly to the reviewers, I suggest that the authors add more motivating examples and intuitions. I support acceptance of this paper.